# OpenReview forum: "Securing Multimodal Large Language Models: Defending Against Jailbreak Attacks with Adversarial Tuning"
_ICLR.cc/2025/Conference — ICLR 2025 Conference Withdrawn Submission_

### Official Review · Reviewer_w3um · 2024-10-18

**Soundness:** 3
**Presentation:** 2
**Contribution:** 1
**Rating:** 3
**Confidence:** 5

**Summary:**

This paper introduces SAFEMLLM, a novel adversarial tuning framework to defend multimodal large language models (MLLMs) against jailbreak attacks. The framework operates in two stages: 1) generating adversarial perturbations using a contrastive embedding attack (CoE-Attack), and 2) updating model parameters to neutralize perturbation effects while preserving model utility. The authors evaluate SAFEMLLM across six MLLMs and six jailbreak methods spanning multiple modalities, demonstrating its effectiveness in defending against diverse attacks without compromising normal interactions.

**Strengths:**

- Comprehensive experiments are conducted across multiple models and jailbreak attack methods, demonstrating the effectiveness of this framework.
- The idea is easy to understand and the paper is easy to follow.

**Weaknesses:**

- This paper lacks novelty. Though it's the first paper to perform adversarial tuning on MLLMs, the attack method is only conducted on the embedding layer and there is no specific modification to the components directly related to VLLM such as the image encoder. Previous work[1, 2 ] on LLMs has already demonstrated the effectiveness of adversarial training in the latent space. Meanwhile, previous work also propose to first perform attacks on multiple layers of LLM (including the embedding layer) and then fine-tune the model against such attacks. Since this method only performs attacks on the embedding layer, it's not clear why previous work cannot be directly applied to this task.
-  More recent attack methods could be added, such as GPTFuzzer [3] and PAIR [4].


[1] Sheshadri, Abhay, et al. "Targeted latent adversarial training improves robustness to persistent harmful behaviors in llms." arXiv preprint arXiv:2407.15549 (2024).

[2] Casper, Stephen, et al. "Defending Against Unforeseen Failure Modes with Latent Adversarial Training." arXiv preprint arXiv:2403.05030 (2024).

[3] Jiahao Yu, Xingwei Lin, and Xinyu Xing. 2023. Gpt-fuzzer: Red teaming large language models with auto-generated jailbreak prompts. arXiv preprint arXiv:2309.10253

[4] Patrick Chao, Alexander Robey, Edgar Dobriban, Hamed Hassani, George J Pappas, and Eric Wong. 2023. Jailbreaking black box large language models in twenty queries. arXiv preprint arXiv:2310.08419.

**Questions:**

- How is this method different from other latent adversarial training methods [1] [2] ?
- Could this training approach be adapted to improve model safety against other types of attacks?



[1] Sheshadri, Abhay, et al. "Targeted latent adversarial training improves robustness to persistent harmful behaviors in llms." arXiv preprint arXiv:2407.15549 (2024).

[2] Casper, Stephen, et al. "Defending Against Unforeseen Failure Modes with Latent Adversarial Training." arXiv preprint arXiv:2403.05030 (2024).

---

> ### Author Response · Authors · 2024-11-21
> **Rebuttal by Authors (Part 1/2)**
>
> Thank you for your valuable suggestions. Our responses are provided below, and we hope they can address your concerns.\
> `>>> Weakness 1 and Question 1`: ***This paper lacks novelty. Though it's the first paper to perform adversarial tuning on MLLMs, the attack method is only conducted on the embedding layer and there is no specific modification to the components directly related to VLLM such as the image encoder. Previous work[1, 2 ] on LLMs has already demonstrated the effectiveness of adversarial training in the latent space. Meanwhile, previous work also proposes to first perform attacks on multiple layers of LLM (including the embedding layer) and then fine-tune the model against such attacks. Since this method only performs attacks on the embedding layer, it's not clear why previous work cannot be directly applied to this task. How is this method different from other latent adversarial training methods [1] [2] ?***\
> `>>> Response`: Although SafeMLLM and recently proposed latent adversarial training (LAT) methods both adopt an adversarial training framework, we assert that they still differ significantly from SafeMLLM in the following aspects:
>
> (1) ***Different Perturbation Sets***: Unlike jailbreak attacks on LLMs that are based on discrete text tokens, attackers on MLLMs can directly inject unrestricted pixel-level perturbations into images to compromise safety alignment. Thus, we leverage this unique property and propose optimizing unbounded perturbations at the token embedding level, unifying potential adversarial perturbations across different modality inputs in the worst scenario.
>
> However, existing LAT methods [1,2] may not be suitable for our problem. Specifically, when a toxic prompt is used as a training sample, these LAT methods generate noise by adding perturbations to the intermediate token representations corresponding to this text prompt. This design is based on the hypothesis that injecting perturbations into the intermediate text token features is equivalent to, and even stronger than, adding extra non-word adversarial text tokens after the query in terms of attack intensity. However, this hypothesis may ***no longer stand for MLLM attacks as attackers not only inject perturbations through discrete text tokens but can also introduce noise via images with continuous values.*** More importantly, in MLLMs, the number of tokens for images largely exceeds that of text prompt tokens (e.g. 576 tokens on LLaVA-13B), making these latent adversarial training methods ***less effective.***
>
> Thus, this design is novel and different from LAT-based method. We also conduct an experiment to validate our model design by comparing the proposed SafeMLLM with the lastest LAT-based method [2]. Specifically, we follow the setting used in [2], and inject perturbations into latent token features with $\epsilon=20$ in the L2 constraint. For a fair comparison, we adopted the same optimization target in SafeMLLM and also increased $\epsilon$ to 40 to explore scenarios under stronger attacks. The modified method is named SafeMLLM-LAT. The results of using the ImgJP attack method on LLaVA-13B are shown in the table below:
>
>    || SafeMLLM-LAT($\epsilon=20$)| SafeMLLM-LAT($\epsilon=40$)| SafeMLLM|
>    |-| :-: | :-: |:-:|
>    |ASR|    13.00 | 12.00 |0.00 |
>
> As shown in the above table, adding perturbations to the latent representations performs worse than directly using the adversarial token embeddings, which confirms our hypothesis.
>
> (2) ***Different training objectives***. Besides the perturbation sets, another difference lies in the training objectives in both the attack loop and model updating step. The first LAT method [1] sets up the adversarial training in an untargeted manner.  The second LAT method [2] improves this into a targeted attack objective, which optimizes the adversarial noise on the affirmative response and updates the model based on another rejective response.
>
> However, our attack objective differs from the above two approaches. ***In addition to the target loss, we propose a contrastive loss that adaptively suppresses the probabilities of generating unexpected texts at different steps.*** During the attack optimization, the proposed loss enhances the perturbation strength by reducing the probability of sampling the rejective response. Correspondingly, during the model updating process, the contrastive loss improves the model's robustness by increasing the probability difference between sampling the safety and toxic responses. The experiments in Section 5.2 and Table 2 further verify this point, showing that the inclusion of contrastive loss significantly improves ASR performance across three MLLMs.
>
> Overall, our proposed SafeMLLM is different from the LLM-based LAT methods in the above aspects. The experimental results in Table 1 also demonstrate the effectiveness of our proposed adversarial training algorithm.

---

> > ### Comment · Reviewer_w3um · 2024-11-21
> > **Official Comment by Reviewer  w3um**
> >
> > First, thank you very much for the clarification on the other questions.
> >
> > Second, I agree with the intuition of the paper that latent adversarial training is a natural approach for enhancing MLLM safety since attackers on MLLMs can directly inject unrestricted pixel-level perturbations into images to compromise safety alignment. However this still doesn't answer my question below:
> >
> > **Meanwhile, previous work also propose to first perform attacks on multiple layers of LLM (including the embedding layer) and then fine-tune the model against such attacks. Since this method only performs attacks on the embedding layer, it's not clear why previous work cannot be directly applied to this task.**
> >
> > Your follow-up experiment focuses on attack on the embedding layer with untargeted noise, however since images are treated as input embeddings after being extracted with vision-extractor and projectors, they should function similarly to the latent of other text tokens in later layers. Under this circumstance, attacking multiple layers of LLM on the overall latent composed of both image latents and text latents could potentially yield better performance. In paper[1], they have demonstrated that perturbing the residual stream at multiple layers rather than a single layer, each with its own ε constraint, typically yields better results. Could you please clarify this issue and conduct additional experiments based on targeted adversarial training with multiple layers as described in paper [1]?
> >
> > Meanwhile, there is also an incorrect citation in your response: **The first LAT method [1] sets up the adversarial training in an untargeted manner. The second LAT method [2] improves this into a targeted attack objective.**
> >
> > [1] Sheshadri, Abhay, et al. "Targeted latent adversarial training improves robustness to persistent harmful behaviors in llms." arXiv preprint arXiv:2407.15549 (2024).
> >
> > [2] Casper, Stephen, et al. "Defending Against Unforeseen Failure Modes with Latent Adversarial Training." arXiv preprint arXiv:2403.05030 (2024).

---

> > > ### Author Response · Authors · 2024-11-21
> > >
> > > Thanks for your response. We sincerely apologize for the confusion caused by the reversed order of the two references in our rebuttal, where [1] refers to the latest LAT attack method.
> > >
> > > We also want to clarify the response to ***Weakness 1 and Question 1***. In our new experiment, We did follow your suggestion and follow [1] to inject perturbations. Precisely, we adhered to their settings by injecting noise into the multiple layers, including five layers: ['embedding', '8', '16', '24', '30']. The results are shown in the table above.  Although both SafeMLLM and the LAT method [1] inject perturbations into the embedding layer,  we optimize perturbation on additional tokens rather than limiting them to the token embeddings of the toxic query prompt only in [1]. Considering the attacker can leverage a large number of additional tokens introduced by adversarial image noise to execute the attack in MLLMs, our design better aligns with this unique property and can thus perform better.
> > >
> > > We hope our clarification can adequately address your concerns regarding the novelty of our work. Your invaluable comments make our contributions clearer and more distinguishable from existing work.

---

> ### Author Response · Authors · 2024-11-21
> **Rebuttal by Authors (Part 2/2)**
>
> `>>> Weakness 2`: ***More recent attack methods could be added, such as GPTFuzzer [3] and PAIR [4]***\
> `>>> Response`: Thanks for pointing this out. We have added descriptions of GPTFuzzer [3] and PAIR [4] into the related work section (page 3, lines 151-155) of the revised manuscript. We have also conducted an experiment by using PAIR to attack the LLaVA-13B model. For the experiment setup, we follow PAIR [4] to use 50 samples from AdvBench, and set Vicuna-13B-v1.5 as the attacker's  LLM. The number of query times is set to 20. The results are illustrated in the following table:
>
> ||Original|VLGuard|R2D2|CAT|SafeMLLM|
>  |-| :-: | :-: |:-:| :-: |:-:|
> |ASR|38.00  |20.00 |16.00 | 12.00|0.00  |
>
> From the table, we can observe that our proposed SafeMLLM still outperforms other baselines, which demonstrates the effectiveness of our proposed method in defending against such jailbreak attacks.
>
> `>>> Question 2`: ***Could this training approach be adapted to improve model safety against other types of attacks?***\
> `>>> Response`: Although our proposed SafeMLLM is targeted for MLLM jailbreak attacks, it can also be adapted to other adversarial attack threats. As mentioned in the existing work [a], there is another adversarial threat for MLLMs, where the attacker optimizes an adversarial image to make the MLLM output a designated text string $\alpha$, regardless of the user’s input queries. The text string contains a specific phishing link aimed at luring users into a scam (e.g., “View more details at https://phishinglink!”).
>
> We adapted SafeMLLM for this specific string attack by replacing the malicious query dataset in Algorithm 1 with a manually curated dataset. To create this dataset, we first sample 100 normal queries from the Alpaca training set [b]. For each normal query,  we create $c_n$ by prompting gpt-4-turbo to generate diverse templates with an HTTP prefix (e.g., "Download the details at https:"). We directly use the ground truth answer from the dataset as the negative response $r_n$, and keep the other parts of Algorithm 1 unchanged to train the target MLLM.
>
> After adversarial training, we follow [a] to use stochastic gradient descent to optimize an adversarial image with the goal of making the MLLM output the string $\alpha$. Note that the string  $\alpha$ and phishing link are unknown during the adversarial training. We follow [a] to conduct the optimization on the Alpaca training set with another 520 samples, and we evaluate the ASR on 100 held-out queries from the same dataset.
>
> The results based on LLaVA-13B are shown in the table below. SafeMLLM still significantly outperforms the original model under this adversarial threat, demonstrating our proposed method's overall generalization capability.
>
> ||Original|SafeMLLM|
>  |-| :-: |:-:|
> |ASR|88.00  |2.00
>
>
> [a] Bailey, L., Ong, E., Russell, S., & Emmons, S. Image Hijacks: Adversarial Images can Control Generative Models at Runtime. ICML 2024 \
> [b] Taori, R., Gulrajani, I., Zhang, T., Dubois, Y., Li, X., Guestrin, C., Liang, P., and Hashimoto, T. B. Stanford alpaca: An instruction-following llama model. https://github.com/tatsu-lab/ stanford_alpaca, 2023.]

---

> ### Comment · Reviewer_w3um · 2024-11-22
> **Official Comment by Reviewer w3um**
>
> Thanks for your clarification. This is a friendly reminder that your response to Reviewer 16HR contains the same citation error.
>
> I remain unconvinced about the novelty of perturbing both image and text tokens for two main reasons:
>
> - In [1], the authors only attack token embeddings of toxic query prompts because this aligns with their specific objectives (they chose not to attack the conversation template tokens). Since image tokens are integral to the response generation in your case, extending their method to include image embeddings would be a natural idea. However, your follow-up experiments do not explore this direction.
> - The LLaVA architecture only treats images differently during the initial feature extraction and projection stages. After that, image embeddings are concatenated with text embeddings in the embedding layer and processed identically. **Therefore, applying the existing LAT method [1] to VLLMs would only require adjusting the attack indices which is a very small modification, as both image and text representations are fundamentally treated the same way**.
>
> Additionally, you haven't addressed a crucial question: Would attacking both image and text embeddings across multiple VLLM layers improve performance? If such an approach yields better results, it would suggest that **a minor modification to existing methods** could outperform your proposed approach.
>
>
>
> [1] Sheshadri, Abhay, et al. "Targeted latent adversarial training improves robustness to persistent harmful behaviors in llms." arXiv preprint arXiv:2407.15549 (2024).

---

> ### Comment · Reviewer_zCNu · 2024-11-22
> **Comment on the discussion**
>
> To give another perspective:
>
> Reviewer w3um22 remarks that the contribution is not significant. However, in the same regard, [1] is taking the same algorithm as [2] but explores more than just the first latent layers (a trivial contribution?). Moreover, both works explore continuous attacks in the latent space, which have already been explored in computer vision and other domains before LLMs (trivial again?). Long story short, nearly every work in ML can be viewed as "not novel" or "trivial".
>
> The proposed work is, to the best of my knowledge, the first to conduct a study on adversarial training in VLMs with arguably some interesting takeaways for the community. Besides this contribution, the authors evaluate different and novel loss functions.
>
> I am not convinced that proposing a more complicated loss / framework would strengthen the contribution.
>
> [1] Sophie Xhonneux, Alessandro Sordoni, Stephan G¨unnemann, Gauthier Gidel, and Leo Schwinn. Efficient adversarial training in llms with continuous attacks. arXiv preprint arXiv:2405.15589, 2024
>
> [2] Sheshadri, Abhay, et al. "Targeted latent adversarial training improves robustness to persistent harmful behaviors in llms." arXiv preprint arXiv:2407.15549 (2024).

---

> > ### Comment · Reviewer_w3um · 2024-11-22
> > **Official Comment by Reviewer w3um**
> >
> > Thanks for your comment. Let me systematically address the novelty concerns:
> >
> > 1 .Regarding publication timing:
> > - Paper [1] was published on May 24, 2024
> > - Paper [2] was published on July 22, 2024
> > - Being within a 3-month period, these would be considered part of the same submission cycle. In such case, if both papers were submitted to the same conference, they could reasonably claim concurrent development of latent adversarial training for LLMs.
> > - Moreover, neither paper[1,2] has been published at any academic conference to my knowledge. If paper [1] was already public since May 2024, and paper[2] made an submission to ICLR in October, the novelty requirement would be substantially higher since [1] would constitute prior work.
> >
> > 2. Regarding Novelty of this paper
> > - As I mentioned in my previous review: **Therefore, applying the existing LAT method [2] to VLLMs would only require adjusting the attack indices which is a very small modification, as both image and text representations are fundamentally treated the same way in the backbone LLM.**  Additionally, authors haven't addressed a crucial question: Would attacking both image and text embeddings across multiple VLLM layers improve performance? If such an approach yields better results, it would suggest that a minor modification to existing methods could outperform their proposed approach.
> >
> > - To make my views clearer, consider these two analogous cases:
> >
> >      - Case 1: Previous work proposes latent adversarial training in traditional computer vision models[3], and follow-up work transfers LAT to LLMs[1,2] - this represents a significant novel contribution.
> >
> >      - Case 2: Previous work proposes a novel training method for computer vision classification models, and follow-up work simply applies this method to the backbone of an object detection model claiming better performance - this represents an incremental adaptation.
> >
> > This work aligns more with Case 2, representing a straightforward application of an existing technique rather than a substantial technical advancement. Therefore, I believe the current work lacks sufficient novelty for ICLR.
> >
> > [1] Sophie Xhonneux, Alessandro Sordoni, Stephan G¨unnemann, Gauthier Gidel, and Leo Schwinn. Efficient adversarial training in llms with continuous attacks. arXiv preprint arXiv:2405.15589, 2024
> >
> > [2] Sheshadri, Abhay, et al. "Targeted latent adversarial training improves robustness to persistent harmful behaviors in llms." arXiv preprint arXiv:2407.15549 (2024).
> >
> > [3] Park, Geon Yeong, and Sang Wan Lee. "Reliably fast adversarial training via latent adversarial perturbation." Proceedings of the IEEE/CVF international conference on computer vision. 2021.

---

> ### Author Response · Authors · 2024-11-23
> **Response to Reviewer w3um (Part 1/2)**
>
> ***Dear Reviewer w3um:***
>
> ***We will respond to your question from the following two aspects:***
>
> ***1. When transferring LAT methods into your problem, would attacking both image and text embeddings across multiple VLLM layers improve performance?***\
> We follow your comments and use the LAT method [a] to attack image and text token embeddings across multiple MLLM layers. The jailbreak method is ImgJP attack, the dataset is AdvBench, and the victim MLLM is LLaVA-13B. Here, we still set the intermediate attacking layers as ['embedding', '8', '16', '24', '30'] following [a], and the image token embeddings are obtained from a random image at each training iteration. This method is named as ***LAT(img+txt)***. The results are shown in the table below, and we also compare the average runtime per adversarial training iteration.
>
> ||runtime (sec)|ASR|
>  |:-:|:-:|:-:|
> |LAT(img+txt)|192.39  |3.00|
> |SafeMLLM|38.70  |0.00 |
>
> **Performance Comparison**: From the result, we can observe that the ASR of LAT(img+txt) is ***worse*** than that of our proposed SafeMLLM, although it outperforms other baselines, such as R2D2 [b] and CAT [c]. The reason is that in MLLMs, a single image often corresponds to a large number of tokens. For instance, in LLaVA-13B, an image is represented by 576 tokens. Considering that the noise needs to be injected into an excessive number of tokens across multiple MLLM layers, it could increase the difficulty of perturbation optimization, potentially leading to issues such as overfitting on targeted affirmative responses. This, in turn, impacts the corresponding model updates during the defense process. In our experiments, we also observe this phenomenon by extending the token numbers from $P_0^h$ and $P_0^t$ in the hyperparameter sensitivity analysis (page 19, lines 1005-1011).
>
> One potential solution to this issue is to attack only a subset of image tokens on a subset of intermediate layers. However, it is less practical in the LAT setting. This is because it requires determining ***the number of attack tokens***, ***the positions of these tokens in the image***, and ***the specific layers*** to target across various MLLMs, which presents a series of very tricky ablations. In contrast, our proposed SafeMLLM only requires setting the number of tokens from $P_0^h$.
>
> ***Efficiency Comparison***: More importantly, our proposed SafeMLLM is almost ***five times faster*** than LAT(img+txt) in terms of runtime, even though both methods perform the same number of optimization steps in the attack loop. We also attribute this to the large number of image tokens in MLLMs. During adversarial training, these numerous tokens need to go through multiple forward passes in the attack loop, significantly increasing computational resources. However, SafeMLLM only leverages 8 tokens, thus making it more efficient. As a result, we believe this experiment can demonstrate the effectiveness of our proposed perturbation design.
>
> [a] Sheshadri, Abhay, et al. Targeted latent adversarial training improves robustness to persistent harmful behaviors in llms. arXiv 2024.
>
> [b] Mazeika et al. Harmbench: A standardized evaluation framework for automated red teaming and robust refusal. ICML 2024.
>
> [c] Xhonneux et al. Efficient adversarial training in llms with continuous attacks. arXiv 2024.

---

> > ### Author Response · Authors · 2024-11-23
> > **Response to Reviewer w3um (Part 2/2)**
> >
> > ***2. This work aligns more with Case 2, representing a straightforward application of an existing technique rather than a substantial technical advancement. Therefore, I believe the current work lacks sufficient novelty for ICLR.***
> >
> > We respectfully disagree with your point here. Firstly, this paper represents ***the first work*** on defending against jailbreak attacks specifically targeting MLLMs. It is important to note that ***MLLMs and LLMs exhibit significant differences*** when it comes to the threats posed by jailbreak attacks. As described in the paper (lines 89-91), attackers can leverage multiple modalities to inject perturbations into MLLMs, which creates unique challenges that our work aims to address.
> >
> > To tackle this issue, we are ***the first to introduce new token embeddings ${P_h^0,P_t^0}$ at specific positions in the prompt query***. By injecting perturbations into these new embeddings, we effectively unify adversarial noise across different modalities. We do not extend the LAT method by injecting perturbed noise into image and text tokens to avoid computationally intensive optimization on a large number of token embeddings. Our experiments clearly demonstrate the superiority of this novel design.
> >
> > Additionally, we ***introduce a new training objective*** that applies to both the attack and defense phases, further enhancing the robustness of our methodology. The effectiveness of this objective is validated through comprehensive experiments.
> >
> > ***Thank you once again for your constructive comments. We kindly ask you to reconsider the novelty of our work and adjust your rating accordingly. We believe our work goes beyond a straightforward application of existing techniques. Specifically, the proposed SafeMLLM model demonstrates both greater effectiveness and efficiency compared to intuitive solutions.***

---

> > > ### Author Response · Authors · 2024-11-25
> > >
> > > Dear Reviewer w3um,
> > >
> > > Thank you for the multi-round discussions. We have conducted additional experiments to further demonstrate the effectiveness of the proposed SafeMLLM and highlighted its superior efficiency compared to the method you suggested. We would like to emphasize that our approach introduces a novel methodology rather than a straightforward application of existing techniques. Details can be found in the above replies.
> > >
> > > As the rebuttal period nears its end, we are still awaiting your response and hope that our clarifications have sufficiently addressed your concerns regarding the novelty of our work.
> > >
> > > Best,
> > >
> > > Authors

---

> > ### Comment · Reviewer_w3um · 2024-11-25
> > **Official Comment by Reviewer w3um**
> >
> > Thank you for providing additional experimental results. After careful consideration, I maintain my original review and rating due to limited novelty in the work. I will explain my concerns from two main aspects:
> >
> > - Performance Analysis
> >     - The performance improvement is modest, showing only a 3% increase in Attack Success Rate (ASR) compared to the previous LAT method [1]. This is particularly noteworthy given that the previous LAT approach [1] already outperforms all other baselines on LLaVA demonstrated in your original paper.
> >
> >     - Regarding efficiency, while efforts to reduce computational costs in adversarial training have been valuable in traditional computer vision [2,3], I question its relevance as a performance metric for LLM adversarial training. This is reflected in your original paper, where efficiency comparisons were relegated to the Appendix with limited experiment.
> >
> >     -  Though you suggest that achieving the 3% improvement requires complex ablation experiments like attacking subset image tokens or selecting target layers. However, since the previous method was designed for traditional LLMs without extensive hyperparameter optimization for follow-up experiment, similar improvements might be achievable through basic parameter tuning (e.g., learning rate adjustments). This is especially relevant given that your method's performance also depends significantly on the hyperparameter λ shown in Appendix I.
> >
> > - Inituion
> >     - As I have mentioned in my previous review, **the LLaVA architecture only treats images differently during the initial feature extraction and projection stages. After that, image embeddings are concatenated with text embeddings in the embedding layer and processed identically.**  Since the backbone LLM processes image and text embeddings uniformly and there is no specific optimization exists for image embeddings,  I fail to understand how the image embeddings differ fundamentally from the latent representations defined in [1], which raises questions about the novelty of adapting LAT from LLMs to VLLMs.
> >    - The small **3%** performance gap between your method and previous LAT work [1] (which wasn't specifically designed for image attacks) supports this view. As noted in my previous review, **the existing method [1] with minor modifications to attack indices selection** could likely achieve comparable performance.
> >
> > Consequently, I will maintain my score for this paper. Thanks again for your thorough experimental results.
> >
> > [1] Sheshadri, Abhay, et al. Targeted latent adversarial training improves robustness to persistent harmful behaviors in llms. arXiv 2024.
> >
> > [2] Shafahi, Ali, et al. "Adversarial training for free!." Advances in neural information processing systems 32 (2019).
> >
> > [3] Wong, Eric, Leslie Rice, and J. Zico Kolter. "Fast is better than free: Revisiting adversarial training." arXiv preprint arXiv:2001.03994 (2020).

---

> ### Author Response · Authors · 2024-12-03
>
> Dear reviewer w3um:
>
> We would like to point out your biased evaluation regarding our contributions:
>
> ***1. Misunderstanding the proposed SafeMLLM.*** We must stress that we ***DO NOT*** adapt LAT from LLMs to VLLMs. We are puzzled by the continued perception of our work as merely a straightforward adaptation of LAT, despite our repeated clarifications about the unique contributions of our paper. ***These include a completely different training objective and distinct perturbation sets.*** The only similarity between LAT and SafeMLLM lies in their shared use of perturbations injected into the embedding layer—an approach common across various previous image attack methods that perturb image inputs. Additionally, we have demonstrated that the LAT method is suboptimal in our problem setting, particularly due to its inefficiency. This underscores the necessity of the SafeMLLM framework, which is fundamentally different from LAT, offering superior ASR performance while maintaining computational efficiency.
>
> ***2. Inconsistencies in your reviews.*** Initially, you stated: “If such an approach yields better results, it would suggest that a minor modification to existing methods could outperform their proposed approach.” However, after we conducted the experiment and demonstrated the superior performance of SafeMLLM, the focus shifted to criticizing the performance improvement as “marginal.” We would like to clarify the reasoning behind this so-called “marginal” improvement. The experiment was conducted on LLaVA-13B using the ImgJP Attack. As shown in Table 1, LLaVA-13B itself exhibits a notable degree of robustness, with the most effective baseline method, CAT [a], achieving an ASR of only 4%. In this context, the 3% performance improvement achieved by SafeMLLM should ***NOT*** be considered marginal. Additionally, due to time constraints during the rebuttal period, we were unable to conduct experiments on all MLLMs and attack methods. Nonetheless, we believe this trend will be corroborated across more models and attack methods, particularly those where SafeMLLM demonstrates substantial improvements over the baseline in ASR performance.
>
> ***3. Efficiency in LLM adversarial training.*** Efficiency is a critical consideration in LLM adversarial training, and we strongly disagree with your assertion otherwise. Efficiency is a fundamental concern for most machine learning tasks, as it directly impacts the feasibility of implementing algorithms in real-world scenarios. Adversarial training is no exception to this. For instance, the latest LLM adversarial training method, CAT [a], recently accepted at NeurIPS 2024, explicitly focuses on addressing efficiency challenges. Given this context, it is evident that efficiency must be a priority in the adversarial training process, as it significantly affects the practicality and scalability of the approach. Ignoring this aspect undermines the relevance and applicability of the proposed methods.
>
> ***4. Limited experimental results on efficiency.***  As stated in our methodology section (lines 207–209), using an entire image during adversarial training can significantly impact computational efficiency. To verify this, we conducted an experiment presented in Appendix H. This observation also applies to the LAT(img+txt) method, which similarly introduces noise to an entire image during adversarial training, leading to the same efficiency concerns. It is important to note that we did not report efficiency metrics in the main paper because none of the baseline methods in Table 1 utilize the image during adversarial training; thus, their computational efficiency is comparable. In conclusion, as discussed both above and in Appendix H, the efficiency issue is an unavoidable challenge when incorporating the entire image into adversarial training. However, SafeMLLM addresses this concern effectively, achieving a significant computational advantage—it is five times faster than LAT(img+txt).
>
> Based on the points outlined above, ***characterizing our work as a straightforward application of an existing technique (e.g., LAT) is neither accurate nor reflective of the methodological innovations and contributions we have introduced.***
>
> While we fully respect your role in evaluating the contributions of the papers you review, we, as authors, hope our work is assessed fairly and accurately. It is important to recognize that building a researcher’s reputation is significantly more challenging than rejecting a paper. We ask that our contributions be considered in the broader context of advancing the field, and we trust that the evaluation process upholds a fair and constructive standard.
>
> [a] Xhonneux et al. Efficient adversarial training in llms with continuous attacks. NeurIPS 2024.

---

### Official Review · Reviewer_zCNu · 2024-10-25

**Soundness:** 4
**Presentation:** 3
**Contribution:** 3
**Rating:** 5
**Confidence:** 4

**Summary:**

The authors propose the first adversarial training algorithm for VLMs. They introduce a new loss function based on contrastive learning for attack optimization and model training and demonstrate the effectiveness of the new components in an ablation study. Compared to prev. work in the LLM setting, they show that their algorithm provides higher robustness.

**Strengths:**

* Multimodal adversarial training is an underexplored research area and this paper provides numerous insights on some of the design choices that need to be considered in this context.
* Robustness is evaluated from multiple perspectives, which makes it easier to assess differences between multi-modal and LLM only robustification approaches.
* State-of-the-art attack methods are used for evaluation
* Exhaustive hyperparameter ablations are conducted for the presented method.

**Weaknesses:**

* Doesnt the argument in line 90 only apply to the method proposed by Mazeika et al.? I think the framing could be improved.
* The two-step algorithm is just standard adversarial training with nontypical loss functions. I think it would be easier to understand if the authors framed their contribution accordingly. Initially, I thought that this 2 stage algorithm would be something different.
* Include the dataset that was used for the experiment in Figure 3
* “We attribute this to the fact that in all MLLMs, the image is always placed before the text as input” This information would be helpful when you describe the method. To give a better intuition about P_0^h

**Questions:**

* Do I understand it correctly, that no images are used during attack optimization? If yes, did you ablate if encoding a given image and starting the attack from this initialization is helpful?
* What are the hyperparameters of the competitor approaches? Are there any ablations regarding the “best” or reasonably “best” possible achievable robustness with the different methods. Hard to compare methods directly to each other otherwise
* Could the authors clarify how they measured over-refusal after safety training in their framework?

In summary I believe this to be an interesting contribution to the ICLR and robustness community and would be willing to increase my score if all my concerns and questions are addressed.

---

> ### Author Response · Authors · 2024-11-21
> **Rebuttal by Authors (Part 1/2)**
>
> Thank you for the valuable review. We hope the following responses can address your concerns.\
> `>>>Weakness 1`: ***Doesnt the argument in line 90 only apply to the method proposed by Mazeika et al.? I think the framing could be improved.***\
> `>>>Response`: Thank you for noting this. The argument in line 90 is not limited to R2D2 [1] but also applies to other LLM-based adversarial training methods that optimize adversarial noise on the discrete text tokens, such as the method proposed in [2]. We have noticed that the citation for the related work was missing here, and we have corrected the omission and expression in the revised version (page 2, line 91-94).
>
> [1] Mantas Mazeika, Long Phan, Xuwang Yin, Andy Zou, Zifan Wang, Norman Mu, Elham Sakhaee, Nathaniel Li, Steven Basart, Bo Li, David A. Forsyth, and Dan Hendrycks. Harmbench: A standardized evaluation framework for automated red teaming and robust refusal. In ICML. OpenReview.net, 2024.\
> [2] Liu, F., Xu, Z., & Liu, H. (2024). Adversarial tuning: Defending against jailbreak attacks for llms. arXiv preprint arXiv:2406.06622.
>
> `>>>Weakness 2`: ***The two-step algorithm is just standard adversarial training with nontypical loss functions. I think it would be easier to understand if the authors framed their contribution accordingly. Initially, I thought that this 2 stage algorithm would be something different.***\
> `>>>Response`: Thanks for your advice. We have modified the related descriptions in the Abstract section (page 1, lines 18-25),  where we originally used "two-stage" to describe SafeMLLM, which could be misunderstood. Also,  we have added a paragraph summarizing this paper's contributions at the end of the introduction section (page 3, lines 125-134). We hope this better highlights the unique contributions and effectiveness of our work.
>
> `>>> Weakness 3`: ***Include the dataset that was used for the experiment in Figure 3***\
> `>>> Response`:  Thanks for pointing this out. As described in Section 5.2, we adopt 100 samples from the LLaVA-Instruct-80K dataset for the utility experiment in Figure 3. We have added this information in the caption of Figure 3 (page 9, lines 439-442) for clarity and better understanding.
>
> `>>> Weakness 4`: ***We attribute this to the fact that in all MLLMs, the image is always placed before the text as input” This information would be helpful when you describe the method. To give a better intuition about $P_0^h$***\
> `>>> Response`: Thank you for your valuable suggestions. We have added this explanation in the revised manuscript (page 3, lines 247-251 ). We believe this addition provides better intuition for understanding our method.
>
> `>>> Question 1`: ***Do I understand it correctly, that no images are used during attack optimization? If yes, did you ablate if encoding a given image and starting the attack from this initialization is helpful?***\
> `>>> Response`: Yes, you are correct! No images are used during the attack optimization, and we directly optimize perturbations on the token embeddings {$P_0^h,P_0^t$}.
>
> We also added an ablation experiment to evaluate the effectiveness of attacking a given image instead of embedding the token during the attack. Specifically, we replace the front token embedding $P_0^h$​ with a given image input $​I_0$ and optimize the perturbations on both $I_0$  and the token embedding $P_0^t$ placed after the query in Step I. In Step II, we update the model based on the optimized perturbations accordingly. We refer to this approach as w/ Adv.Image. The experiments are conducted on the LLaVA-7B and LLaVA-13B models using the ImgJP attack, and the results are presented in the table below. In addition to ASR performance, we also measured the average runtime per iteration and GPU memory usage. For all evaluation metrics, lower values indicate better performance.
>
> |LLaVA-7B| runtime (s)↓| GPU Memory (MB)↓| ASR|
>  |-| - | - |-|
> |w/ Adv.Image|84.42  | 32869 |5.00 |
> |SafeMLLM|20.73  | 30291 |6.00 |
>
> |LLaVA-13B| runtime (s)↓| GPU Memory (MB)↓| ASR|
>  |-| - | - |-|
> |w/ Adv.Image|263.56  | 66092 |0.00 |
> |SafeMLLM|38.70  | 57475 |0.00 |
>
> As shown in the table, optimizing image perturbations significantly impacts computational efficiency but does not yield noticeable gains in ASR performance. As a result, we directly attack the token embeddings during the attack optimization. We have added this experiment in Appendix H (page 18, lines 962-971).

---

> ### Author Response · Authors · 2024-11-21
> **Rebuttal by Authors (Part 2/2)**
>
> `>>> Question 2`: ***What are the hyperparameters of the competitor approaches? Are there any ablations regarding the “best” or reasonably “best” possible achievable robustness with the different methods. Hard to compare methods directly to each other otherwise***\
> `>>> Response`: We compare three different methods in our experiment, including VLGuard [1]，R2D2 [2] and CAT [3]. For VLGuard, we do not use any hyperparameters and directly adopt the models officially trained and released by authors [1]. For LLM-based adversarial training methods R2D2 and CAT, we adapt them to our problem by only fine-tuning the LLM decoder from different MLLMs, and we adopt the same hyperparameters as specified in their original implementations. We believe that directly using their original hyperparameters will not significantly impact the ASR values, as the original LLMs targeted in their experiments share the same architecture and model size as the LLM decoders used in our task.
>
> We also analyze the effect of using different hyperparameters on CAT, using ***attack iterations*** as an example. The experiments are conducted on LLaVA-7B using ImgJP, and the results are shown below. The CAT method originally performed ***10*** attack iterations.
>
> |Iterations| 5| ***10***| 20|SafeMLLM|
>  |-| :-: | :-: |:-:|:-:|
> |ASR| 15.00 | ***9.00*** |10.00 |6.00|
>
> As illustrated in the table, adopting different ***attack iterations*** for the baseline method CAT does not improve the ASR performance in our problem setting. Also, considering that performing ablations on different hyperparameters for each model and attack setting would be highly resource-consuming, we thus followed the default settings as outlined in their original paper.
>
> [1] Yongshuo Zong, Ondrej Bohdal, Tingyang Yu, Yongxin Yang, and Timothy M. Hospedales. Safety fine-tuning at (almost) no cost: A baseline for vision large language models. In ICML. OpenReview.net, 2024 \
> [2] Mantas Mazeika, Long Phan, Xuwang Yin, Andy Zou, Zifan Wang, Norman Mu, Elham Sakhaee, Nathaniel Li, Steven Basart, Bo Li, David A. Forsyth, and Dan Hendrycks. Harmbench: A standardized evaluation framework for automated red teaming and robust refusal. In ICML. OpenReview.net, 2024. \
> [3] Sophie Xhonneux, Alessandro Sordoni, Stephan G¨unnemann, Gauthier Gidel, and Leo Schwinn. Efficient adversarial training in llms with continuous attacks. arXiv preprint arXiv:2405.15589, 2024
>
> `>>> Question 3`: ***Could the authors clarify how they measured over-refusal after safety training in their framework?***\
> `>>> Response`: We measure the over-refusal via the utility evaluation as described in Section 5.2, where we adopt 100 benign image-text questions from LLaVA-Instruct-80K and follow LLaVA to use GPT-4 score each model’s responses. The results in Figure 3 show that our method can retain utility for these benign questions. A refused response to any benign question will receive a very low score, and we have put some results and their GPT scores in Table 7 on page 20.  Note that all of these samples are extracted from the results when omitting the utility loss in SafeMLLM, which we have also discussed in the qualitative analysis ( lines 1022-1025, page 19).
>
>  To further evaluate the utility of SafeMLLM, we also add an experiment on a widely-used MLLM evaluation benchmark-MM-Vet [1]. The benchmark contains 217 multimodal questions and adopts gpt-4-turbo to evaluate the responses from the following dimensions:  Recognize (Rec), OCR, Knowledge (Know), Language Generation (Gen), Spatial awareness (Spat), and Math. The results on LLaVA-7B and LLaVA-13B are reported in the table below:
>
> | LLaVA-7B| rec  | ocr  | know | gen  | spat | math | total |
> |-------------|------|------|------|------|------|------|-------|
> |  Original   | 36.9 | 24.0 | 18.5 | 20.5 | 28.0 | 3.8  | 32.2  |
> | VLGuard| 33.9 | 22.9 | 13.8 | 14.2 | 27.2 | 3.8  | 30.1  |
> | R2D2| 34.7 | 21.5 | 16.4 | 18.1 | 24.3 | 7.7  | 30.2  |
> | CAT| 37.7 | 20.1 | 24.3 | 25.1 | 25.7 | 3.8  | 31.5  |
> | SafeMLLM| 37.5 | 24.1 | 20.5 | 21.1 | 28.3 | 3.8  | 32.5  |
>
> | LLaVA-13B| rec  | ocr  | know | gen  | spat | math | total |
> |-------------|------|------|------|------|------|------|-------|
> | ori         | 42.1 | 25.9 | 24.4 | 25.1 | 30.4 | 11.2 | 36.0  |
> | vlguard     | 37.7 | 26.6 | 17.7 | 21.4 | 30.9 |  3.8 | 32.9  |
> | R2D2        | 41.1 | 26.2 | 24.4 | 26.1 | 32.0 |  7.7 | 35.4  |
> | RTEAT       | 42.7 | 27.7 | 26.7 | 26.1 | 32.7 | 15.0 | 36.9  |
> | safemlllm   | 44.0 | 27.1 | 23.8 | 25.6 | 34.0 | 15.0 | 37.8  |
>
> For each metric, higher values indicate better performance. We observe that SafeMLLM maintains response quality across all aspects, further demonstrating that our method does not compromise the overall capabilities of the target MLLM. We add this experiment in Appendix G (page 18, lines 949-956).
>
> [1] Yu, W., Yang, Z., Li, L., Wang, J., Lin, K., Liu, Z., ... & Wang, L. MM-Vet: Evaluating Large Multimodal Models for Integrated Capabilities.ICML 2024.

---

> > ### Comment · Reviewer_zCNu · 2024-11-21
> > **Thank you for the response**
> >
> > I want to thank the authors for their extensive response and the additional experiments they conducted. My concerns were sufficiently addressed.
> >
> > I will follow the discussions with the other reviewers and raise my score depending on the outcome. Currently, I am leaning toward accepting the paper.

---

> > > ### Author Response · Authors · 2024-11-21
> > >
> > > Thank you for your prompt response. We’re glad to hear that our replies have adequately addressed your concerns. We also appreciate your positive support for our work and hope you might consider raising your ratings following the rebuttal.

---

> > > > ### Author Response · Authors · 2024-11-23
> > > > **Thank you for Discussing with Reviewer w3um**
> > > >
> > > > We thank you once again for supporting our work and acknowledging the novelty and significance of our contributions.

---

> > > > > ### Comment · Reviewer_zCNu · 2024-11-25
> > > > > **Final decision**
> > > > >
> > > > > Sorry for the delay. I have considered the rebuttal of the other reviewers, changes in the paper, and relevant related work.
> > > > > I adjusted my score based on the discussions to slightly below the acceptance threshold. Ultimately, I agree that the advantage compared to existing methods is unclear.

---

> > > > > > ### Author Response · Authors · 2024-12-03
> > > > > >
> > > > > > Dear reviewer zCNu:
> > > > > >
> > > > > > We regret that your perspective has shifted during the rebuttal period. Nevertheless, we would like to reiterate our contributions and highlight the advantages of our work over existing methods from the following three perspectives:
> > > > > >
> > > > > > ***Pioneering Work on Defending Against Jailbreak Attacks on MLLMs***\
> > > > > > This paper is the first to address the specific challenge of defending Multimodal Large Language Models (MLLMs) against jailbreak attacks. It is crucial to recognize that MLLMs and LLMs face fundamentally different threats from these attacks. As outlined in the paper, attackers can exploit multiple modalities to inject perturbations into MLLMs, creating unique challenges that our work seeks to address.
> > > > > >
> > > > > > ***Innovative Approach to Unified Adversarial Noise Injection***\
> > > > > > To tackle these challenges, we propose introducing new token embeddings, $(P_h^0, P_t^0)$, at specific positions in the prompt query. By injecting perturbations into these embeddings, we unify adversarial noise across modalities. Importantly, this approach avoids the computational overhead associated with injecting perturbed noise directly into image and text tokens, as seen in LAT. This design results in a solution that is significantly more efficient and effective. Our experiments demonstrate that SafeMLLM not only outperforms LAT across various dimensions but is also five times faster in computational efficiency. While we could not test all MLLMs and attack methods during the rebuttal period due to time constraints, we believe these results will generalize, particularly for cases where SafeMLLM shows substantial improvements in ASR performance.
> > > > > >
> > > > > > ***Distinct Training Objective for Robust Defense***\
> > > > > > We introduce a completely new training objective applicable to both the attack and defense phases, enhancing the robustness of our methodology. This novel objective has been validated through comprehensive experiments, further distinguishing our work from existing approaches.
> > > > > >
> > > > > > We strongly disagree with reviewer ***w3um’s*** characterization of our work as an extension of LAT. As demonstrated through rigorous experiments, our proposed SafeMLLM is a more refined and effective solution for defending against jailbreak attacks on MLLMs.

---

### Official Review · Reviewer_16HR · 2024-10-29

**Soundness:** 3
**Presentation:** 2
**Contribution:** 3
**Rating:** 6
**Confidence:** 4

**Summary:**

This paper presents a novel adversarial training method called SAFEMLLM, aimed at strengthening VLM models against jailbreak attacks. Specifically, SAFEMLLM consists of two phases: (1) introducing adversarial embedding tokens, where adversarial tokens are initialized in the token embedding layer and optimized to produce adversarial responses; (2) using the adversarial noise from Phase 1 and additional defense data to fine-tune the model for robustness. The authors validate the effectiveness of the proposed defense method across six mainstream VLM model architectures.

**Strengths:**

- This work addresses the security defense of multimodal VLMs, which is crucial for enhancing the secure deployment of VLMs.
- Comprehensive experiments conducted on multiple model architectures and datasets demonstrate the effectiveness of the proposed method.

**Weaknesses:**

- The definitions of formulas are unclear. For example, what do $P_h$ and $P_t$ represent in lines 240-244? Why initialize two tokens instead of one?

- Why does the CoE-attack optimize at the embedding token level? Why not at the representation level?


- How does $\text{equ.2}$ guarantee output quality while ensuring the model’s replies are relevant to the query?

- There is formula redundancy. Both equ.3 and equ.4} describe the same optimization objective, yet the authors define it twice as $loss_{adv}$ in equ.3 and as $loss_{def}$ in equ.4. This complexity should be reduced; maintaining clear formula definitions is essential to avoid redundancy.

- The results in Table 3 are unclear. The authors state that results were tested on ImgJP (image) and AdvBench (text) datasets, yet the table does not present the experimental results for these two modalities separately. Additionally, what is the distinction between $J_{target}/J_{contra}$ and $L_{target}/J_{contra}$? The authors should avoid using multiple ambiguous definitions, as this complicates understanding the experimental results.


- What are the specific settings for model output quality in Figure 3? The authors tested only 100 samples, which may not effectively reflect changes in model output capabilities. It is recommended that the authors test on 500 benign samples to assess the effects accurately.

- How can the model avoid generating garbled outputs? In lines 511-513, the authors indicate that using $L_{contra}$ and $J_{contra}$ to fine-tune the model may lead to meaningless garbled outputs (e.g., repeating "safe"). I request the authors to further explain the reasons for this phenomenon and how to prevent garbled outputs during optimization. This is crucial for assessing the effectiveness of the optimization steps in their method. Additionally, could the authors provide examples of model outputs in both successful and failed optimization scenarios?

Overall, I acknowledge the proposed adversarial training method for enhancing the robustness of VLMs against jailbreak attacks. However, there are still issues regarding unclear loss definitions and ambiguous optimization details in the methods section. On the other hand, in the experimental section, the authors need to further elucidate the roles of the different loss components. Moreover, an explanation and experimental results on how to prevent the model from generating meaningless outputs (such as repetitive and low-quality responses) during adversarial training are also required. If the authors can address the aforementioned concerns, I would like to increase my score.

**Questions:**

see weakness.

---

> ### Author Response · Authors · 2024-11-21
> **Rebuttal by Authors (Part 1/4)**
>
> Thank you for the valuable review. We have provided our responses below. We hope these responses can address your concerns.\
> `>>> Weakness 1`: ***The definitions of formulas are unclear. For example, what do $\mathbf{P}_0^h$ and $\mathbf{P}_0^h$ represent in lines 240-244? Why initialize two tokens instead of one?***\
> `>>>Response`:  We apologize for the vague clarification here. $\mathbf{P}_0^h$ and $\mathbf{P}_0^t$ are two matrices initialized from word token embeddings. Each matrix has a shape of $K\times C$, where $K$ is the number of tokens and $C$ is the embedding dimension. During the attack optimization, we position the first embedding matrix $\mathbf{P}_0^h$ before the text query to act as the ***adversarial image*** $\mathbf{I}^{\prime}$. This design is based on the fact that in all MLLMs, the image is always placed before the text as input. Similarly, another embedding matrix $\mathbf{P}_0^t$  is positioned after the text query to act as the ***adversarial string suffix***. Therefore, we need two token embedding matrices here. We have rewritten this part in our revised manuscript, please refer to lines 247-251 on page 5 for more details.
>
> `>>> Weakness 2`:***Why does the CoE-attack optimize at the embedding token level? Why not at the representation level?***\
> `>>> Response`: As described in lines 247–251 on page 5, we optimize the perturbation based on two token embeddings: one is placed before the toxic query, and the other is placed after the query. This is a heuristic design as attackers always inject adversarial perturbations from the input level, such as placing an adversarial image in front of the prompt or using an adversarial string suffix. Thus, optimizing perturbations at the token embedding level can unify attacks from both modalities.
>
> We also notice that some existing works inject perturbations into latent representations of Large Language Models (LLMs) [1,2]. Specifically, when a toxic prompt is used as a training sample, these latent adversarial training (LAT) methods generate noise by adding perturbations to the intermediate token representations corresponding to this text prompt. This design is based on the hypothesis that injecting perturbations into the intermediate text token features is equivalent to, and even stronger than, adding extra non-word adversarial text tokens after the query in terms of attack intensity. However, this hypothesis may ***no longer stand for MLLM attacks as attackers not only inject perturbations through discrete text tokens but can also introduce noise via images with continuous values.*** More importantly, in MLLMs, the number of tokens for images largely exceeds that of text prompt tokens (e.g., 576 tokens on LLaVA-13B), making these latent adversarial training methods ***less effective***.
>
> We conducted an extra experiment to validate our intuition. Specifically, we follow the setting in the latest LAT method [1], and inject perturbations into latent token features with $\epsilon=20$ in the L2 constraint. For a fair comparison, we adopted ***the same optimization target*** in SafeMLLM and also increased $\epsilon$ to 40 to explore scenarios under stronger attacks. The modified method is named as SafeMLLM-LAT. The results of using the ImgJP attack method on LLaVA-13B are shown in the table below:
>
> || SafeMLLM-LAT($\epsilon=20$)| SafeMLLM-LAT($\epsilon=40$)| SafeMLLM|
>  |-| :-: | :-: |:-:|
> |ASR|    13.00 | 12.00 |0.00 |
>
> As shown in the above table, adding perturbations to the latent representations performs worse than directly using the adversarial token embeddings, which validates our hypothesis.
>
> [1] Sheshadri, Abhay, et al. "Targeted latent adversarial training improves robustness to persistent harmful behaviors in llms." arXiv preprint arXiv:2407.15549 (2024).
>
> [2] Casper, Stephen, et al. "Defending Against Unforeseen Failure Modes with Latent Adversarial Training." arXiv preprint arXiv:2403.05030 (2024).
>
> `>>> Weakness 3`: ***How does the contrastive loss in Eq.2 guarantee output quality while ensuring the model’s replies are relevant to the query?***\
> `>>> Response`: The contrastive loss proposed here is designed to suppress the log probability of the model producing unexpected responses, thereby enhancing the effectiveness of the defense. Only using the constrastive loss during adversarial training cannot ensure that the outputs after attacks or model updates remain relevant to the query, and we put more detailed discussions in response to Weakness 7.

---

> ### Author Response · Authors · 2024-11-21
> **Rebuttal by Authors (Part 2/4)**
>
> `>>> Weakness 4`: ***There is formula redundancy. Both equ.3 and equ.4} describe the same optimization objective, yet the authors define it twice as loss_adv in Eq.3 and as loss_def in Eq.4. This complexity should be reduced; maintaining clear formula definitions is essential to avoid redundancy.***\
> `>>> Response`: We want to emphasize that although $L_{adv}$ (Eq. 3) and $L_{def}$ (Eq. 4) share similar expressions, ***they have different optimization targets and variables.*** Specifically, we adopt $L_{adv}$ to optimize the perturbation matrices {$P_0^h, P_0^t$} by increasing the probability of sampling the affirmative response $c_n$​ and decreasing the probability of sampling the refusal response $r_n​$. Conversely, $L_{def}$ is used to update model parameters by increasing the probability of sampling $r_n​$​ and decreasing the probability of sampling $c_n$​​. Therefore, we adopt different notations here.
>
> `>>> Weakness 5`: ***The results in Table 2 are unclear. The authors state that results were tested on ImgJP (image) and AdvBench (text) datasets, yet the table does not present the experimental results for these two modalities separately. Additionally, what is the distinction between  $J_{target}$/$J_{contra}$ and  $L_{target}$/$L_{contra}$? The authors should avoid using multiple ambiguous definitions, as this complicates understanding the experimental results.***\
> `>>> Response`: Thanks for pointing this out, and we are sorry for the vague clarification in the caption of Table 2. The robustness experiments in Table 2 are developed using the ImgJP attack method. We follow the original setup in ImgJP to conduct experiments on the AdvBench datasets, where ImgJP optimizes an adversarial image to make MLLM output affirmative responses for queries in the AdvBench dataset. We have rewritten this part in the caption of Table 2 for clear understanding.
>
> In the original manuscript,  $L_{target}$ is the target loss defined in Eq.1, and $L_{contra}$ is the contrastive loss defined in Eq.2 . They are used as the attack objective for updating adversarial perturbations. In the model updating step, we originally adopted $J_{target}$ and  $J_{contra}$ to represent the target loss and contrastive loss for optimizing the model parameters, and they are defined in Eq.4.  We apologize for any confusion in the notation definitions.  In the revised version, we use $L_{adv}^{target}$ and $L_{adv}^{contra}$ to denote the target and contrastive loss during the attack step, respectively.  Similarly, We adopt $L_{def}^{target}$ and $L_{def}^{contra}$ to denote the target and contrastive loss during the model updating step, respectively. We have also rewritten the description in the caption of Table 2 (page 9, lines 443-451) for better understanding.

---

> ### Author Response · Authors · 2024-11-21
> **Rebuttal by Authors (Part 3/4)**
>
> `>>> Weakness 6`: ***What are the specific settings for model output quality in Figure 3? The authors tested only 100 samples, which may not effectively reflect changes in model output capabilities. It is recommended that the authors test on 500 benign samples to assess the effects accurately.***\
> `>>> Response': Thanks for your valuable suggestions. For the experimental setup in Figure 3, we follow LLaVA by adopting gpt-4-turbo and using the same prompt to score each model’s responses on a scale from 1 to 10.  We report the average score on 100 samples extracted from the LLaVA-Instruct-80K dataset.
>
> ***We have expanded the utility experiments to more comprehensively evaluate the impact of SafeMLLM on the model's general capabilities via two extra evaluations on 517 samples.***
>
> First, in addition to the original 100 samples used in Figure 3, we additionally sampled 200 samples from the LLaVA-Instruct-80K dataset and adopted the same method to score each model’s responses based on gpt-4-turbo. We conducted experiments on both LLaVA-7B and LLaVA-13B, and the average scores over 300 test samples (100+200) are shown below:
>
> |LLaVA-7B| Original | VLGuard | R2D2| CAT| SafeMLLM|
>  |-| - | - |-|-|-|
> |ASR|7.65  | 7.67 |7.58 |7.62|7.64|
>
> |LLaVA-13B| Original | VLGuard | R2D2| CAT| SafeMLLM|
>  |-| - | - |-|-|-|
> |ASR|7.79  | 7.73 |7.68 |7.54|7.73|
>
> From the table, we can observe that after training the model with SafeMLLM, its response quality on these benign questions has not been moderately affected.
>
> We also adopt MM-Vet [1], a widely-used MLLM evaluation benchmark, to comprehensively evaluate the capability of SafeMLLM across various aspects. The benchmark contains 217 multimodal questions and adopts gpt-4-turbo to evaluate the target model’s responses from the following dimensions:  Recognize (Rec), OCR, Knowledge (Know), Language Generation (Gen), Spatial awareness (Spat), and Math. The results on LLaVA-7B and LLaVA-13B are reported in the table below. For each metric, higher values indicate better performance.
>
> | LLaVA-7B| rec  | ocr  | know | gen  | spat | math | total |
> |-------------|------|------|------|------|------|------|-------|
> |  Original   | 36.9 | 24.0 | 18.5 | 20.5 | 28.0 | 3.8  | 32.2  |
> | VLGuard| 33.9 | 22.9 | 13.8 | 14.2 | 27.2 | 3.8  | 30.1  |
> | R2D2| 34.7 | 21.5 | 16.4 | 18.1 | 24.3 | 7.7  | 30.2  |
> | CAT| 37.7 | 20.1 | 24.3 | 25.1 | 25.7 | 3.8  | 31.5  |
> | SafeMLLM| 37.5 | 24.1 | 20.5 | 21.1 | 28.3 | 3.8  | 32.5  |
>
> | LLaVA-13B| rec  | ocr  | know | gen  | spat | math | total |
> |-------------|------|------|------|------|------|------|-------|
> | ori         | 42.1 | 25.9 | 24.4 | 25.1 | 30.4 | 11.2 | 36.0  |
> | vlguard     | 37.7 | 26.6 | 17.7 | 21.4 | 30.9 |  3.8 | 32.9  |
> | R2D2        | 41.1 | 26.2 | 24.4 | 26.1 | 32.0 |  7.7 | 35.4  |
> | RTEAT       | 42.7 | 27.7 | 26.7 | 26.1 | 32.7 | 15.0 | 36.9  |
> | safemlllm   | 44.0 | 27.1 | 23.8 | 25.6 | 34.0 | 15.0 | 37.8  |
>
> From the table, we observe that SafeMLLM still maintains response quality across all aspects. Finally, based on these two experiments involving more than 500 image-text questions (300+217), we demonstrate that SafeMLLM minimally compromises the overall capabilities of the target MLLM. We have added both experiments in Appendix G (page 17, lines 908-917; page 18, lines 949-956).
>
> [1] Yu, W., Yang, Z., Li, L., Wang, J., Lin, K., Liu, Z., ... & Wang, L. MM-Vet: Evaluating Large Multimodal Models for Integrated Capabilities.ICML 2024.

---

> ### Author Response · Authors · 2024-11-21
> **Rebuttal by Authors (Part 4/4)**
>
> `>>> Weakness 7`: ***How can the model avoid generating garbled outputs? In lines 511-513, the authors indicate that using L_contra and J_contra to fine-tune the model may lead to meaningless garbled outputs (e.g., repeating "safe"). I request the authors to further explain the reasons for this phenomenon and how to prevent garbled outputs during optimization. This is crucial for assessing the effectiveness of the optimization steps in their method. Additionally, could the authors provide examples of model outputs in both successful and failed optimization scenarios?***\
> `>>> Response`: We need to point this out that using the contrastive loss (Eq.2) in SafeMLLM does not output garbled texts during adversarial training. Instead, we find that when only using the contrastive loss as the optimization target in the attack step and model updating step, such as setting $L_{adv}=L_{adv}^{contra}$ and $L_{def}=L_{def}^{contra}$, the model may produce meaningless texts based on the toxic training samples with optimized perturbations, even after the model parameters are updated in Step II. This occurs because, ***when only using the contrastive loss in training, the model merely amplifies the probability difference between sampling the refusal response $r_n$​ and the affirmative response $c_n$​.*** This does not guarantee an increase in the probability of generating $r_n$, the expected output after updating the model. Consequently, the model may produce garbled outputs, thereby reducing the effectiveness of adversarial training.
>
> To further verify this claim, we design an experiment by plotting the average negative log probabilities of generating affirmative label $c_n$ and rejective label $r_n$ during the attack optimization steps at different model training iterations (i=1, 50, 100, 150, 200), as illustrated in Figure 4. From Figures 4(d) and 4(e), we observe that for the method that uses only the contrastive loss as the optimization target (dashed line),  the probabilities of the model generating $c_n$ and $r_n$​ are both very low after the training is converged, even though their difference remains large.
>
> SafeMLLM addresses this issue by combining the contrastive loss with the target loss in Eq. 3 and Eq. 4. The target loss acts as a supervised term, guiding the model to generate the expected outputs at different steps. This claim can also be verified in Figures 4(d) and 4(e). When combining the target loss in training (solid line), the model can output the refusal response $r_n$ with a noticeably high probability, regardless of the optimized perturbations. This demonstrates that the model can effectively counteract the introduced perturbations and increase its robustness against jailbreak attacks.
>
>
> ***Additionally, could the authors provide examples of model outputs in both successful and failed optimization scenarios?***
>
> Thanks for your advice. We have provided additional qualitative analysis (page 19, lines 1015-1021) in our case study, please refer to Appendix J.

---

> > ### Comment · Reviewer_16HR · 2024-11-23
> >
> > Thanks for the response, which addressed most of the concerns. After reviewing the revised paper and considering the feedback from other reviewers, I have decided to increase my score to 6.

---

> > > ### Author Response · Authors · 2024-11-23
> > >
> > > Thank you very much for raising your rating. We sincerely appreciate your insightful comments, which have been invaluable in helping us enhance the quality of our work.

---

### Official Review · Reviewer_nQhb · 2024-11-04

**Soundness:** 3
**Presentation:** 3
**Contribution:** 3
**Rating:** 6
**Confidence:** 4

**Summary:**

This paper presents a novel defense framework, SAFEMLLM, that employs the CoE-Attack strategy to craft adversarial embeddings and iteratively refines model parameters, ensuring resilience against attacks while preserving benign input performance. Comprehensive experiments across six multimodal language models and six jailbreak attack techniques validate SAFEMLLM's effectiveness, especially in challenging white-box scenarios.

**Strengths:**

1. This paper proposes a novel adversarial training framework, SAFEMLLM, and demonstrates its effectiveness across diverse MLLMs and jailbreak methods in white-box scenarios, showcasing robust defense without sacrificing user interaction
2. This paper excels in its writing style, which is fluid, clear, and easy to read.
3. This paper stands out with thorough experimental comparisons across three jailbreak attack scenarios: image-based, text-based, and image-text jailbreak attacks.

**Weaknesses:**

1. The Introduction could be strengthened by more clearly emphasizing the article's unique contribution, originality, and effectiveness. Currently, these elements are not sufficiently highlighted, which might undermine the article's impact.
2. The proposed framework has limitations as it only considers image and text data, excluding audio, video, and other multimedia types.
3. Authors fail to provide experimental results demonstrating that their framework can reduce overall computing resources.

**Questions:**

1. In Section 4, four datasets were chosen for robustness evaluation. Please explain why the paper selected there four datasets? Please provide criteria and rationale for their authority and suitability. For the LLaVA-Instruct-80K dataset, specifically, why was it chosen to assess the utility of the fine-tuned models? A brief summary of each dataset's relevance in the main text would also enhance readability.
2. The introduction part presents previous work, such as VLGuard, which is effective against black-box attacks but fails to defend against white-box attacks. Please provide further explanations on why this occurs. Specifically, how does an attacker's possession of parameters and gradient information enable them to launch more effective attacks in white-box scenarios, and what are the key points of defense in such cases.
3. Figure 2 presents an overview of the proposed SAFEMLLM framework, with a focus on the first phase, CoE-Attack. In this phase, the paper emphasizes the optimization of two noise metrics: noisy image and noisy text. Are these the only types of noise considered in the SAFEMLLM, or does the framework have the potential to incorporate other modalities such as video and audio? If not, it it better to discuss the limitations of the current approach in terms of its applicability to a broader range of multimodal data?
4. Why are there many missing values for the VLGurad metric in the performance indicators listed in Table 1?

---

> ### Author Response · Authors · 2024-11-21
> **Rebuttal by Authors (Part 1/2)**
>
> Thank you for your thoughtful review. We have provided our responses below.\
> `>>> Weakness 1`: ***The Introduction could be strengthened by more clearly emphasizing the article's unique contribution, originality, and effectiveness. Currently, these elements are not sufficiently highlighted, which might undermine the article's impact.***\
> `>>> Response`: Thanks for your suggestion. We added a paragraph (page 3, lines 125-133) to summarize this paper's contributions at the end of the introduction section, including an overview of the research problem, the proposed methodology, and the experimental results. We hope this highlights the unique contributions and effectiveness of our work.
>
> `>>> Question 1`: ***In Section 4, four datasets were chosen for robustness evaluation. Please explain why the paper selected there four datasets? Please provide criteria and rationale for their authority and suitability. For the LLaVA-Instruct-80K dataset, specifically, why was it chosen to assess the utility of the fine-tuned models? A brief summary of each dataset's relevance in the main text would also enhance readability.***\
> `>>> Response`:  In our experiment, we evaluated six attack methods on four jailbreak datasets. We ***follow the original paper*** to use the same implementations and dataset for robustness evaluation for each attack method, ensuring that the hyperparameters used in the attack setup are optimal. For the utility evaluation, we follow LLaVA [1] and adopt the LLaVA-Instruct-80K dataset, using gpt-4-turbo to evaluate the model's responses. We chose this dataset because it contains diverse and complex instruction-following multimodal VQA samples to assess the MLLM's comprehension and generation abilities across various scenarios.
>
> We apologize for the lack of clarity regarding dataset usage in Section 4, and we have added a brief summary of each dataset’s relevance in the main text (page 7, lines 332-339).
>
> [1] Haotian Liu, Chunyuan Li, Qingyang Wu, and Yong Jae Lee. Visual instruction tuning. In NeurIPS, 2023a
>
> `>>> Question 2`: ***The introduction part presents previous work, such as VLGuard, which is effective against black-box attacks but fails to defend against white-box attacks. Please provide further explanations on why this occurs. Specifically, how does an attacker's possession of parameters and gradient information enable them to launch more effective attacks in white-box scenarios, and what are the key points of defense in such cases.***\
> `>>> Response`: VLGuard enhances model safety by performing supervised fine-tuning on a safety instruction-following dataset. The dataset contains diverse harmful information presented through images, text, or multimodal inputs, with refusal responses as labels. In ***black-box*** scenarios, attackers cannot access the model's parameters and directly place harmful queries in the image or text inputs, such as injecting malicious text requests into the input image[1,2]. Therefore, safety-trained MLLMs like VLGuard have already used such data during training, enabling the models to output safe responses.
>
> However, in ***white-box*** scenarios, attackers can compromise safety alignment by injecting adversarial noise, where the "adversarial noise" refers to trainable parameters applied to the inputs, such as adjusting pixel values. To create a more effective attack, attackers can optimize these parameters with an objective that enforces the model to output an affirmative response, such as, "Sure, here are the steps to create a bomb." Since MLLMs autoregressively generate subsequent content based on previous outputs, they are more likely to continue in an affirmative tone and produce detailed harmful content. Therefore, compared to manually crafted prompts in black-box attacks, white-box attacks can leverage these adversarial perturbations to explicitly shift the model's output space, resulting in more effective attacks.
>
> Safety-trained MLLMs, such as VLGuard, are no longer effective in white-box scenarios, as these adversarial perturbations are not included in the training data of these methods. As a result, the key to defending against white-box jailbreak attacks on MLLMs is to reduce the risk of attackers identifying such adversarial noise that can alter the model's safety outputs.
>
> [1] Xin Liu, Yichen Zhu, Jindong Gu, Yunshi Lan, Chao Yang, and Yu Qiao. Mm-safetybench:
> A benchmark for safety evaluation of multimodal large language models. arXiv preprint
> arXiv:2311.17600, 2023b.
>
> [2] Yichen Gong, Delong Ran, Jinyuan Liu, Conglei Wang, Tianshuo Cong, Anyu Wang, Sisi Duan, and Xiaoyun Wang. Figstep: Jailbreaking large vision-language models via typographic visual prompts. arXiv preprint arXiv:2311.05608, 2023

---

> ### Author Response · Authors · 2024-11-21
> **Rebuttal by Authors (Part 2/2)**
>
> `>>> Weakness 2 & Question 3`: ***Figure 2 presents an overview of the proposed SAFEMLLM framework, with a focus on the first phase, CoE-Attack. In this phase, the paper emphasizes the optimization of two noise metrics: noisy image and noisy text. Are these the only types of noise considered in the SAFEMLLM, or does the framework have the potential to incorporate other modalities such as video and audio? If not, it is better to discuss the limitations of the current approach in terms of its applicability to a broader range of multimodal data?***\
> `>>> Response`: Thanks for your comments. We admit that we only consider two commonly used modalities since most existing multimodal jailbreak methods only focus on these two modalities. However, the proposed ***SafeMLLM has the potential to be extended to address potential attacks involving additional modalities.*** For example, for the video-based jailbreak attacks, SafeMLLM could insert more adversarial tokens before the query to counter the adversarial noise across a sequence of multiple image frames. Alternatively, we could interleave perturbed embeddings within the inputs to adapt flexibly to modality switching. As you suggested, we have discussed the limitations regarding SafeMLLM’s applicability to a broader range of multimodal data in the revised manuscript (Appendix L, page 21).
>
> `>>> Weakness 3`: ***Authors fail to provide experimental results demonstrating that their framework can reduce overall computing resources.***\
> `>>> Response`: Thank you for pointing this out. As mentioned in Section 3.2 (page 4, lines 206-208), optimizing an adversarial image in front of the toxic query and an adversarial text string after the query simultaneously in Step I could be highly computationally intensive. To validate this claim, we conducted an additional experiment. Specifically, we replace the front token embedding $\mathbf{P}_0^h$ with a given image input $​I_0$, and optimize the perturbations on both ​$I_0$  and the token embedding $P_0^t$ placed after the query in Step I. In step II, we update the model based on the optimized perturbation accordingly.  We refer to this approach as w/ Adv.Image. We test it against SafeMLLM on the LLaVA-7B and LLaVA-13B models using the ImgJP attack, comparing the average runtime per iteration (step I + step II) and GPU memory usage. The results are illustrated in the table below:
>
> |LLaVA-7B| runtime (s)↓| GPU Memory (MB)↓| ASR↓|
>  |-| - | - |-|
> |w/ Adv.Image|84.42  | 32869 |5.00 |
> |SafeMLLM|20.73  | 30291 |6.00 |
>
> |LLaVA-13B| runtime (s)↓| GPU Memory (MB)↓| ASR↓|
>  |-| - | - |-|
> |w/ Adv.Image|263.56  | 66092 |0.00 |
> |SafeMLLM|38.70  | 57475 |0.00 |
>
> As shown in the table, optimizing image perturbations significantly impacts computational efficiency but does not yield noticeable gains in ASR performance, thereby validating our claim that simultaneously optimizing the perturbations on the original inputs is not practical. We have added this experiment in Appendix H (page 18, lines 962-971).
>
> `>>> Question 4`: ***Why are there many missing values for the VLGurad metric in the performance indicators listed in Table 1?***\
> `>>>Response`: As mentioned in lines 347-350 on page 7, we evaluate VLGuard in Table 1 by using the models officially trained and released by [1] to ensure fairness in comparison. Since ***VLGuard has not released checkpoints for models trained on MiniGPT-4 and InstructBLIP,*** we did not report ASR results for these two types of models.
>
> [1] Yongshuo Zong, Ondrej Bohdal, Tingyang Yu, Yongxin Yang, and Timothy M. Hospedales. Safety fine-tuning at (almost) no cost: A baseline for vision large language models. In ICML. OpenReview.net, 2024

---

> > ### Author Response · Authors · 2024-11-25
> >
> > Dear Reviewer nQhb,
> >
> > Thank you for your insightful and constructive comments. As the rebuttal period nears its end, we are still awaiting your reply and hope that our responses have adequately addressed your concerns.
> >
> > Best,
> >
> > Authors

---

### Note · Authors · 2024-12-12

I have read and agree with the venue's withdrawal policy on behalf of myself and my co-authors.